# Reasons for Initiation and Regular Use of Heated Tobacco Products among Current and Former Smokers in South Korea: Findings from the 2020 ITC Korea Survey

**DOI:** 10.3390/ijerph20064963

**Published:** 2023-03-11

**Authors:** Hong Gwan Seo, Steve S. Xu, Grace Li, Shannon Gravely, Anne C. K. Quah, Sungkyu Lee, Sujin Lim, Sung-il Cho, Yeol Kim, Eon Sook Lee, Yoon-Jung Choi, Connor R. Miller, Maciej L. Goniewicz, Richard J. O’Connor, Geoffrey T. Fong

**Affiliations:** 1Department of Family Medicine, National Cancer Center, Goyang 10408, Republic of Korea; 2Department of Cancer Control, Graduate School of Cancer Science and Policy, National Cancer Center, Goyang 10408, Republic of Korea; 3Department of Psychology, University of Waterloo, Waterloo, ON N2L 3G1, Canada; 4Korea Center for Tobacco Control Research and Education, Seoul 04554, Republic of Korea; 5National Tobacco Control Center, Korea Health Promotion Institute, Seoul 04933, Republic of Korea; 6Graduate School of Public Health, Seoul National University, Seoul 08826, Republic of Korea; 7Department of Family Medicine, Inje University, Ilsanpaik Hospital, Goyang 10380, Republic of Korea; 8Division of Cancer Prevention and Population Sciences, Department of Health Behaviors, Roswell Park, Comprehensive Cancer Center, Buffalo, NY 14263, USA; 9School of Public Health Sciences, University of Waterloo, Waterloo, ON N2L 3G1, Canada; 10Ontario Institute for Cancer Research, Toronto, ON M5G 0A3, Canada

**Keywords:** heated tobacco products (HTPs), South Korea, HTP-cigarette consumers, HTP consumers, reasons

## Abstract

South Korea is the world’s second-largest heated tobacco product (HTP) market after Japan. HTP sales in South Korea have increased rapidly since May 2017, accounting for 10.6% of the total tobacco market in 2020. Despite this, little is known as to why HTP consumers who were current and former smokers started using HTPs and used them regularly. We analyzed cross-sectional data for 1815 adults (aged 19+) who participated in the 2020 International Tobacco Control (ITC) Korea Survey, of whom 1650 were HTP-cigarette consumers (those who reported smoking cigarettes and using HTPs ≥ weekly) and 165 were exclusive HTP consumers (using HTPs ≥ weekly) who were former or occasional smokers (smoking cigarette < weekly). Respondents were asked to report the reason(s) they used HTPs, with 25 possible reasons for HTP-cigarette consumers and 22 for exclusive HTP consumers. The most common reasons for initiating HTP use among all HTP consumers were out of curiosity (58.9%), family and friends use HTPs (45.5%), and they like the HTP technology (35.9%). The most common reasons for regularly using HTPs among all HTP consumers were that they were less smelly than cigarettes (71.3%), HTPs are less harmful to own health than cigarettes (48.6%), and stress reduction (47.4%). Overall, 35.4% of HTP-cigarette consumers reported using HTPs to quit smoking, 14.7% to reduce smoking but not to quit, and 49.7% for other reasons besides quitting or reducing smoking. In conclusion, several common reasons for initiating and regularly using HTPs were endorsed by all HTP consumers who were smoking, had quit smoking completely, or occasionally smoked. Notably, only about one-third of HTP-cigarette consumers said they were using HTPs to quit smoking, suggesting that most had no intention of using HTPs as an aid to quit smoking in South Korea.

## 1. Introduction

Heated tobacco products (HTPs) are tobacco delivery devices that heat processed tobacco (contained in tobacco sticks, plugs, or capsules) to generate an aerosol to be inhaled at a lower temperature (240–350°C) than combustible cigarettes (>600°C) [1]. HTPs contain nicotine and additives, some of which are not found in tobacco [2]. Several studies suggest that both HTP cytotoxicity and emissions of toxic nitrosamines are lower than those of cigarettes, which may benefit smokers if they completely switch to HTPs [3,4,5]. The tobacco industry has indicated that HTPs are a viable and less harmful substitute to cigarettes [6]. However, there is much debate about the relative harm of HTPs compared to cigarettes and whether they are an effective means of helping cigarette smokers with smoking cessation [1]. For example, a Cochrane systematic review that evaluated the effectiveness and safety of HTPs for smoking cessation reported that there were no available studies that have investigated cigarette smoking cessation, so the effectiveness of HTPs for this purpose remains uncertain [7]. There was moderate-certainty evidence that HTP consumers have lower exposure to toxicants, including carcinogens, than cigarette smokers and very low- to moderate-certainty evidence of higher exposure than those attempting to achieve abstinence from all tobacco [7]. Although IQOS devices and heatsticks are currently banned for sale in the United States (US) by the US International Trade Commission, based on evidence that IQOS reduces exposure to harmful chemicals, the US Food and Drug Administration (FDA) granted limited authorization for IQOS systems (including IQOS 3.0) to be marketed as a reduced exposure tobacco product until 7 July 2024 [8,9].

HTPs were introduced in the Republic of Korea (South Korea hereafter) in May 2017 following Japan’s successful national launch of IQOS in 2016 [10]. Tobacco companies have promoted HTPs as a “less harmful” and “cleaner” alternative to cigarettes, claiming they are “free from fire, ash, and smoke” and “without the cigarette-like smell” [11,12]. In 2020, HTP sales accounted for 10.6% of the total tobacco market in South Korea, making it the world’s second-largest market after Japan [13]. Several factors contributed to the rapid rise of HTPs, including increased exposure to them due to the geographic proximity to Japan, high smoking prevalence among Korean adults (particularly males, 42.7% in 2016) [14], and a high propensity for Korean consumers’ preferences for new and innovative products, including nicotine vaping products (NVPs, also known as e-cigarettes) [15]. In 2018, the prevalence of HTP use among Korean adults (used at least once in the past 30 days) was 5.1.% (8.4% of men and 1.5% of women) [16]. About 90% of current HTP consumers were concurrently using at least another type of nicotine product, either cigarettes or NVPs [16].

As the potential harm reduction benefit of HTPs depends on whether they are used to completely substitute cigarettes or help ex-smokers to stay abstinent from cigarettes [17], there is a need to understand the reasons for regular use of HTPs among those who currently smoke or formerly smoked, including assessing which characteristics, features, and/or functions are most important to them. Several industry-independent population studies have examined reasons for using HTPs. The most frequently cited reasons for regular use of HTP among HTP-cigarette and exclusive HTP consumers were that HTPs are less harmful to themselves and others, for personal enjoyment, and have greater social acceptability than cigarettes, according to a large national study in Japan [18]. In a large study conducted in 28 European countries, it was reported that those who use HTPs primarily do so because they are less harmful than smoking, their friends use them, and so they can stop or reduce smoking [19].

To date, there have been three industry-independent studies on reasons for using HTPs in South Korea. According to a qualitative study of 38 Korean adults who currently or formerly used HTPs, reasons for using HTPs were because they are less smelly than cigarettes, to cope with family members’ pressure to quit smoking and to be able to use HTPs in indoor public places where smoking is banned [20]. A cross-sectional survey of 228 young adults (ages 19–24) assessed their awareness, trial, and use of IQOS. Of eight current IQOS consumers (all of whom smoked cigarettes and/or used e-cigarettes), six reported using HTPs because they believed that HTPs are less harmful and less smelly compared to cigarettes, and two reported using IQOS to stop smoking [21]. An online study of 7000 Korean adults, of which 944 (16.8%) had ever used HTPs, found that the most common reasons for using HTPs among ever-HTP consumers were that “no ash is formed” (74.8%) and “it has less odor than conventional cigarettes” (71.9%) [22]. There were limitations to these studies in that they either used small convenience samples or did not provide a comprehensive list of reasons for HTP use.

This study identified the key reasons why HTP consumers who were current, occasional, or former smokers initiated HTP use and why they currently use them regularly. We were interested specifically in whether HTP-cigarette consumers intended to use HTPs for smoking cessation purposes as opposed to for smoking reduction only (without cessation) or only for reasons not related to reducing smoking or for smoking cessation purposes. A secondary analysis examined if there were differences in reasons for HTP use between HTP-cigarette and exclusive HTP consumers.

## 2. Materials and Methods

Cross-sectional data for this study were from Wave 1 of the International Tobacco Control Korea (ITC KRA1) Survey conducted from 18–28 June 2020, a web-based study of 4794 adults (aged 19+) recruited from Rakuten Insight’s web panel. The sample consisted of exclusive cigarette smokers (smoke ≥weekly, but do not use HTP or NVPs ≥weekly), exclusive HTP consumers (use HTPs ≥weekly, but do not smoke or use NVPs ≥weekly), exclusive NVP consumers (use NVPs ≥weekly, but do not smoke or use HTP ≥weekly), dual HTP-cigarette consumers (use both ≥weekly but do not vape ≥weekly), dual NVP-cigarette consumers (use both ≥weekly but do not use HTP ≥weekly), dual HTP-NVP consumers (use both ≥weekly, but do not smoke ≥weekly), triple consumers (use all three ≥weekly), and non-consumers of all three tobacco/nicotine products. The response rate was 15.2%, and the cooperation rate was 97.4%. The weighted data were used in the analysis. These weights were adjusted for unequal selection probabilities and designed to make these 4794 respondents representative of the adult population aged ≥19 years by their nicotine use status. The calibration targets were obtained from the 2019 Korea Community Health Survey, which is the national health survey in Korea. A detailed description of the sample and methods are reported elsewhere [23,24].

Study procedures and materials were reviewed and cleared by the Research Ethics Board, University of Waterloo, Canada (REB# 41512) and the Institutional Review Board of Korean Health Promotion Institute (#120160811107AN01-2004-HR-042-02). All participants provided consent to participate.

### 2.1. Study Sample

Eligible respondents (n = 1815) included in this study were HTP consumers (≥weekly) who were current smokers, occasional smokers, or former smokers. We excluded from this study 62 HTP consumers who were never a smoker in their lifetime. Of 1815 regular HTP consumers, 1650 were regularly smoking cigarettes (1136 dual HTP-cigarette consumers and 514 triple consumers), and 165 were either former smokers or occasional smokers (<weekly). The 165 HTP consumers who were considered exclusive HTP consumers in this study consisted of 115 exclusive HTP consumers who used HTPs ≥weekly but did not currently smoke or use NVPs ≥weekly and 50 dual HTP-NVP consumers who used HTPs and NVPs ≥weekly. From cessation point of view, these 165 exclusive HTP consumers can be further divided into two groups: occasional smokers (n = 117) or former smokers (n = 48) who had stopped smoking completely (26 within the last two years and 22 over two years ago). The study selection process is presented in Appendix A.

### 2.2. Measures

#### 2.2.1. Independent Variables

##### Smoking Status

Respondents were asked: “How often do you currently smoke cigarettes?” Response options included: “daily”, “weekly”, “monthly”, “I have quit smoking completely”, or “I have never smoked”. Those who reported smoking cigarettes daily or weekly and those who had quit smoking were included.

##### HTP Use Status

Respondents were asked: “How often do you currently use HTPs?” Response options included: “daily”, “weekly”, “monthly”, “less than monthly”, or “not at all”. Those who reported using HTPs daily or weekly were included in this study (respondents who used HTPs less than weekly were not considered to be regular HTP consumers in this study).

##### HTP-Cigarette Consumers vs. Exclusive HTP Consumers

Respondents were further classified into ‘HTP-cigarette consumers’ if they smoked cigarettes and were using HTPs ≥weekly (current smokers) or ‘exclusive HTP consumers’ (former or occasional smokers) if they had quit or occasional smoking (<weekly) and were using HTPs ≥weekly.

##### Covariates

Covariates included sex (female vs. male), age groups (19–29, 30–39, 40+), education (high: university degree or higher, moderate: high school or some university, and low: middle school or lower), annual household income (high: KRW 60+ million (approximately >USD 53,000), moderate: KRW 30–60 million (approximately USD 26,500–53,000 USD), low: <KRW 30 million (approximately <USD 26,500, not reported)).

#### 2.2.2. Outcome: Reasons for Initiating and Regular HTP Use

We used the following question to examine reasons for initiating and regularly using HTPs: “Which of the following are reasons that you use heated tobacco products?” with 22 possible reasons for all HTP consumers and 3 extra possible reasons for HTP-cigarette consumers (see Table 1). These reasons are categorized into seven broad themes based on the study conducted in Japan by Xu et al. [18]. The first theme—initiation, covers eight reasons related to the uptake of HTPs. The next five themes—harm reduction, convenience, social consideration, product attractiveness, and personal benefits cover 14 reasons for regularly using HTPs. The last theme—intention, covers three reasons about intentions for regularly using HTPs, but they were only asked of HTP-cigarette consumers.

For each reason, the response options were “yes”, “no”, “refused”, or “don’t know”. Respondents could select multiple reasons for initiating or regularly using HTPs; therefore, these reasons were not mutually exclusive. For analysis purposes, “refused” or “do not know” responses were excluded. Response rates for each item (“yes” or “no”) ranged from 89% for “people in the media or other public figures use HTPs” to 95% for “someone offered me one”. The average response rate for completing all of the items was 93%.

Next, among HTP-cigarette consumers who answered both reasons 23 and 24 (n = 1767) in Table 1, a three-category quit–reduce composite measure of their intentions for using HTPs as per Yong et al. [25] and Gravely et al. [26] to evaluate NVP-use reasons, was derived and coded as: (1) those who selected quitting smoking as a reason for HTP use; (2) those who selected using a HTP to reduce the number of cigarettes they smoke but did not select quitting smoking; or (3) those who did not select reducing the number of cigarettes they smoke or quitting smoking (selected other reasons only).

### 2.3. Statistical Analysis

Sample characteristics were examined using frequencies and unweighted percentages. (Table 2). All other analyses were conducted on weighted data (using calibrated cross-sectional sampling weights). In summary, a raking algorithm was used to calibrate the cross-sectional weights on cigarette smoking status, HTP use status, geographic region, and demographic measures to adjust for the potential disproportional sampling of sub-groups among these categories (such as the oversampling of the dual or triple consumer groups) to make respondents within each of the sub-groups as representative as possible of the corresponding population [27]. The weight calibration was performed using benchmarks from the 2019 Korea Community Health Survey. Further details about weighting can be found elsewhere [23,24].

The overall estimate of each of the possible reasons given by all respondents for initiation and regular use of HTPs came from the separate adjusted logistic regression analyses (using the predicted marginal standardization method (PREDMARG) [28]) adjusted for age group, sex, income, education, and smoking frequency.

The logistic regression analyses were conducted to estimate reasons provided for initiation and regular use of HTPs within each of the two groups (HTP-cigarette vs. exclusive HTP consumer) and to test if there were any differences between them. We also tested the differences between dual HTP-cigarette consumers and triple consumers. General linear contrasts of the predicted marginals in the corresponding models were specified for significance testing of percent differences between HTP-cigarette and exclusive consumers.

Finally, for the composite quit–reduce measure, a multinomial regression analysis was conducted with HTP-cigarette consumers, adjusting for age group, sex, income, education, and smoking frequency. All analyses were conducted SAS callable SUDAAN v11. (SAS Institute Inc. 2013, Cary, NC, USA). All confidence intervals (CIs) and statistical significance were tested at the 95% confidence level.

## 3. Results

### 3.1. Sample Characteristics

Table 2 presents the unweighted sample characteristics of all HTP consumers who were included in this study. The majority of respondents were men (75.1%), had a high education level (86.3%), were currently smoking (90.9%), and smoked cigarettes daily (77.5%). The average age was 41.3.

### 3.2. Reasons Provided for Initiating and Regular Use of HTPs among All HTP Consumers

Table 3 presents the reasons for initiating and regularly using HTPs among all HTP consumers. The three most common reasons for HTP initiation were curiosity (58.9%), family and friends use HTPs (45.5%), and they like the HTP technology (35.9%). The three most common reasons for regular use of HTPs were HTPs are less smelly than cigarettes (71.3%), HTPs are less harmful to own health than cigarettes (48.6%), and stress reduction (47.4%).

### 3.3. Reasons for Initiating and Regular Use of HTPs among HTP-Cigarette Consumers

Among HTP-cigarette consumers, the three most common reasons for initiating HTP use were curiosity (60.8%), family and friends use HTPs (47.5%), they like the HTP technology (35.4%), and attractive heating and charging devices (36.4%). The three most common reasons for regular use of HTPs were HTPs are less smelly than cigarettes (68.8%), HTPs are less harmful to own health (47.6%), and stress reduction (46.3%).

### 3.4. Using HTPs for Smoking Cessation vs. Other Reasons among HTP-Cigarette Consumers

Overall, 36.0% of HTP-cigarette consumers reported using HTPs to quit smoking, 14.7% to reduce smoking but not to quit, and 49.3% for other reasons (not to reduce or quit smoking) (see Table 3-Quit–reduce composite measure).

### 3.5. Reasons for Initiating and Regular Use of HTPs among Exclusive HTP Consumers

Among exclusive HTP consumers, the three most common reasons for initiating HTP use were curiosity (50.3%), likable HTP technology (38.2%), and family and friends using HTPs (36.2%). The three most common reasons for regular use of HTPs were less smelly than cigarettes (83.1%), personal enjoyment (55.3%), and less harmful to own health (54.5%).

### 3.6. Difference in the Reasons for Initiating and Regular Use of HTPs between HTP-Cigarette and Exclusive HTP Consumers

Exclusive HTP consumers were more likely than HTP-cigarette consumers to say that they were using HTPs because HTPs were less smelly than cigarettes (83.1% vs. 68.8%, *p* < 0.01) but less likely to use HTPs because the packaging is attractive (14.4% vs. 26.6%, *p* < 0.01), and because the heating/charging devices are attractive (14.6% vs. 36.2%, *p* < 0.01).

### 3.7. Difference in the Reasons for Initiating and Regular Use of HTPs between Dual HTP-Cigarette and Triple Consumers

As shown in the Appendix A, triple consumers were more likely than dual HTP-cigarette consumers to report that they were using HTPs because of good taste (46.4% vs. 37.7%, *p* = 0.03), the attractive heating/charging devices (44.4% vs. 33.2%, *p* < 0.001); they like the HTP technology (48.1% vs. 31.1%, *p* < 0.001), they are able to use HTPs in places where smoking is banned (42.6% vs. 29.5%, *p* < 0.001), people in the media or other public figures are using HTPs (36.2% vs. 16.8%, *p* < 0.001), attractive packaging (38.1% vs. 22.2%, *p* < 0.001), something to occupy their time (38.9% vs. 20.4%, *p* < 0.001), saving money (31.8% vs. 16.2%, *p* < 0.001), experts like doctors and scientists are using HTPs (29.0% vs. 16.4%, *p* < 0.001), to control their appetite and/or weight (31.9% vs. 15.2%, *p* < 0.001), advice of health professionals to switch to HTPs (27.3% vs. 13.7%, *p* < 0.001), looking cool (23.3% vs. 13.2%, *p* < 0.01), cutting down the number of cigarettes they smoked (43.5% vs. 35.7%, *p* < 0.05), and to replace some of their cigarette while continuing to smoke cigarettes (31.8% vs. 22.6%, *p* < 0.05).

## 4. Discussion

In this study, a nationally representative sample of South Korean HTP consumers who were smoking regularly, not smoking anymore, or smoking only occasionally were examined for the reasons why they initiated and regularly used HTPs. We found that the most common reasons for initiating HTPs were curiosity, family and friends using HTPs, and attractive HTP technology. The most commonly cited reasons for regularly using HTPs were that they are less smelly than cigarettes, stress reduction, enjoyment, being more socially acceptable, and less harmful than cigarettes. This study also shows that there were some differences in reasons for initiating and regularly using HTPs between HTP-cigarette and exclusive HTP consumers.

Marketing for HTPs aims to entice Korean consumers to start using the new generation of non-combustible tobacco products. As in Japan, PMI positioned IQOS as the ‘iPhone’ of tobacco products, opened flagship IQOS stores, and advertised and sold IQOS at CU, the nation’s largest convenience store chain [11,21]. Korea Tobacco & Ginseng Corporation (KT&G) and British American Tobacco (BAT) followed suit in 2018 by launching and marketing their own HTPs, lil and glo, respectively [12,29]. As shown in our study, curiosity and attractive features (e.g., device, technology, packaging) were the most common reasons for initiating HTP use, suggesting the HTP marketing strategies were effective in luring consumers to take up HTPs in the short term.

HTP marketing also focuses on defining HTPs as less smelly and less harmful than cigarettes [11,12,29]. In our study, we found that South Koreans use HTPs primarily for these two reasons. It is interesting that HTP consumers were much more likely to report using HTPs because they were less smelly (71.3%) than because they were less harmful (48.6%). Korean culture considers it appropriate to produce a less offensive smell from smoking since the general public is well aware of the harmful effects of secondhand cigarette smoke [30]. It is possible that social norms of not exposing others to cigarette smoke are more important than reducing personal harm. We also found that 41.8% of all HTP consumers considered HTPs as more acceptable to others, and 31.3% reported that they use HTPs in places where smoking is banned. As almost half of all HTP consumers use HTPs due to their perceived low harm, it is imperative to raise social awareness that HTP emissions are harmful to both consumers and non-consumers [1,4,5].

Less than one-fifth of Korean HTP consumers reported they used HTPs to save costs. Similar to Japan, major tobacco companies are not positioning HTPs as affordable in South Korea [17]. At the time of the survey, the market share of premium cigarette brands (retail price above KRW 4500 (USD 3.70) for 20 sticks/pack) was 5.6%, the share of mid-priced cigarette brands (KRW 4500/pack) was 82.6%, and the share of economy-priced cigarette brands (below KRW 4500/pack) was 11.8% [31]. Almost all HTP tobacco refill brands (PMI’s IQOS heatsticks, BAT’s neosticks, and KT&G’s lil tobacco refills) were sold at the same retail price as mid-priced cigarette brands. Apart from the cost of heat tobacco sticks, HTP consumers also have to pay extra for the HTP devices.

Around 38% of HTP-cigarette consumers used HTPs to reduce their cigarette consumption, and about 35% to quit smoking. Around 25% of HTP-cigarette consumers reported using HTPs to continue smoking. The quit–reduce composite analysis further indicated that around 36% of HTP-cigarette consumers used HTPs to quit smoking, 14.7% to reduce but not quit smoking, and almost half were using HTPs only for reasons other than quitting or reducing smoking. According to these findings, a majority of Korean HTP-cigarette consumers are using HTPs to supplement their cigarette consumption rather than to quit. In the long run, sustained HTP-cigarette use is unlikely to have any positive public health effects (e.g., reducing the harms from smoking or reducing the smoking prevalence) [32,33].

HTP consumers in South Korea commonly use HTPs for the same reason that HTP consumers in Japan do: HTPs are less harmful to their health and others’ health than cigarettes. In Japan, however, the percentage was much higher than that in South Korea. Among all Japanese HTP consumers, 91% used HTPs because they perceived HTPs to be less harmful than smoking cigarettes [17], compared to only 49% among all Korean HTP consumers. In Japan, 86% used HTPs because they perceived HTPs to be less harmful to others, compared to only 42% in South Korea. The government and tobacco industry may have framed the relative risks of HTPs differently in these two countries. In South Korea, both the government and the tobacco industry were involved in the framing of the relative risks of HTPs. Officials from the Korean government stated that HTPs and cigarettes were both equally harmful and that there was no conclusive evidence showing that HTPs were less harmful than cigarettes [34]. In contrast, the tobacco industry claims that HTPs significantly reduce exposure to many toxic chemicals emitted by cigarettes [11,12]. A lack of clear government statement in Japan left consumers primarily exposed to the tobacco industry’s messages about HTPs’ relative harm [35]. Japanese HTP-cigarette consumers were much more likely to use HTPs to help them quit smoking (55.1%) than their Korean counterparts (35.3%) [18]. Japanese HTP-cigarette consumers (64.4%) are also more likely to use HTPs to reduce smoking than Korean HTP-cigarette consumers (38.2%) [18].

Triple consumers of HTPs, NVPs, and cigarettes account for a substantial share of all HTP consumers in South Korea [14,16]. While there is no difference in the most commonly reported reasons for initiating and regularly using HTPs (e.g., curiosity, less smelly, less harmful than cigarettes to own health and the health of others), we found that triple consumers were consistently more likely than dual HTP-cigarette consumers to cite most of the less-commonly reported reasons (e.g., taste good, attractive heating/charging devices, like the HTP technology). The similarities between HTPs and NVPs—both marketed as reduced harm products— coupled with having many other benefits (e.g., less smelly, helped to quit smoking, can use in places where smoking is banned) may account for triple consumers having endorsed a larger variety of less-commonly reported reasons for using HTPs. Moreover, triple consumers may like non-combustible tobacco technology more than dual HTP-cigarette consumers, but this should be further explored.

Among this study’s strengths is that the sample came from a large national sample of the world’s second-largest market for HTP. There are, however, several limitations to consider. First, this is a cross-sectional study; therefore, neither temporality nor causality was investigated between the reasons for HTP use and tobacco-use behavior. Second, exclusive HTP consumers (also former or occasional smokers) were not asked if they were using a HTP to either quit or stay abstinent from cigarette smoking, which would be an important reason to assess for public health reasons. Third, since we report reasons from all HTP consumers using HTPs ≥weekly who were current, occasional, or former smokers, our results may not apply to less frequent HTP consumers or frequent HTP consumers who were never cigarette smokers. Finally, this study was conducted during the early phase of the COVID-19 pandemic. While no lockdown was imposed, several social distancing rules were introduced to slow down the COVID-19 infections, including restricting public gatherings, and mandating masking indoors and outdoors. In South Korea, the pandemic disruption may have affected tobacco-use behaviors and patterns [36].

## 5. Conclusions

Among South Korean adults who currently smoke or formerly smoked cigarettes and who regularly use HTPs, the most common reasons for initiating them were out of curiosity, their family and friend use HTPs, and because of the attractive technology. There were several reasons endorsed for current use, including HTPs being less smelly than cigarettes, for stress reduction, enjoyment, HTPs being more socially acceptable, and being less harmful than cigarettes. Notably, only about one-third of HTP-cigarette consumers said they intended to use HTPs to quit smoking. Thus, HTPs may compromise tobacco control efforts to reduce smoking prevalence if most Korean HTP-cigarette consumers take up HTPs but have no intention to quit cigarette smoking.

## Figures and Tables

**Table 1 ijerph-20-04963-t001:** Reasons for initiating and regularly using HTPs among HTP-cigarette and exclusive HTP consumers.

Theme	Reason
**Initiation of HTPs (#1–8)**	
	Curiosity
	2. Family or friends use HTPs
	3. Experts, like doctors and scientists, use them
Initiation	4. People in the media or other public figures use them
	5. Someone offered me one
	6. I like the technology of heated tobacco products
	7. The heating/charging device is attractive
	8. Health professional advised me to switch to HTP
**Reasons for regular use of HTPs (#9–25)**	
Harm reduction	9. Less harmful to my health than ordinary cigarettes
	10. Less harmful to the health of people around me than ordinary cigarettes
Convenience	11. Use them in places where smoking ordinary cigarettes are banned
Social consideration	12. Make socializing easier
	13. More acceptable than smoking ordinary cigarettes to people around me
Product attractiveness	14. Taste good
	15. The packaging is attractive 16. Do not smell as bad as ordinary cigarettes
Personal benefits	17. I save money by using HTPs instead of smoking ordinary cigarettes
	18. I enjoy using HTPs
	19. Control my appetite and/or weight
	20. Reduce my stress
	21. Makes me look cool
	22. Give me something to do, to occupy my time
Intention *	23. Using them helps me cut down on the number of cigarettes I smoke
	24. Using them might help me stop smoking cigarettes
	25. Replacing some of my ordinary cigarettes with heated tobacco products means I do not have to give up smoking cigarettes altogether

* Asked only of HTP-cigarette consumers.

**Table 2 ijerph-20-04963-t002:** Characteristics of all HTP consumers.

Characteristic	Overall		HTP-Cigarette Consumer		Exclusive HTP Consumer	
	n	%	n	%	n	%
Sex						
Male	1363	75.1	1246	75.5	117	70.9
Female	452	24.9	404	24.4	48	29.1
Age Group						
19–29	265	14.6	237	14.4	28	16.9
30–39	597	32.9	536	32.5	61	37.0
40+	953	52.5	877	53.2	76	46.1
Mean Age	41.3		41.4		41.1	
Education						
Low	14	0.8	10	0.6	4	2.5
Moderate	234	12.9	210	12.8	24	14.7
High	1559	86.3	1424	86.6	135	82.8
Household Income						
Low	31	1.7	26	1.6	5	3.0
Moderate	810	44.6	729	44.2	81	49.1
High	952	52.4	878	53.2	74	44.9
Not Stated	22	1.2	17	1.0	5	3.0
Tobacco Use Status						
Current Smoker	1650	90.9	1650	100.0	0	0.0
Ex- or Occasional Smoker	165	9.1	0	0.0	165	100.0
Smoking Status						
Daily	1407	77.5	1407	85.3	0	0.0
Weekly *	243	13.4	243	14.7	0	0.0
Ex- or Occasional Smoker	165	9.1	0	0.0	165	100.0

Data are unweighted. * Smoke ≥weekly but <daily in the past 30 days.

**Table 3 ijerph-20-04963-t003:** Reasons for initiating and regularly using HTPs among HTP consumers.

Theme/Reason	Overall	HTP-Cigarette Consumer	Exclusive HTP Consumer	Difference (*p*-Value)
	Weighted Percentage (95% Confidence Interval)			
**Personal Benefits: Less Smelly than Ordinary Cigarettes**	**71.3** **(67.8–74.5)**	**68.8** **(65.4–71.9)**	**83.1** **(72.2.3–90.2)**	**0.002**
Initiation: Curiosity	58.9 (54.9–62.8)	60.8 (57.3–64.2)	50.3 (36.2–64.4)	n.s.
Harm reduction: Less harmful to own health	48.6 (44.7–52.6)	47.6 (44.1–51.1)	54.5 (39.4–68.8)	n.s
Personal benefits: Stress reduction	47.4 (43.4–51.4)	46.3 (42.8–49.9)	52.2 (37.5–66.5)	n.s.
Initiation: Family or friends use HTPs	45.5 (41.6–49.5)	47.5 (44.0–51.0)	36.2 (23.8–50.7)	n.s.
Personal benefits: Enjoyment	44.4 (40.6–48.2)	42.4 (38.9–45.9)	55.3 (40.2–69.5)	n.s.
Social consideration: More acceptable to others	41.8 (38.0–45.7)	40.6 (37.1–44.1)	47.7 (34.7–61.0)	n.s.
Harm reduction: Less harmful to others	41.7 (37.8–45.6)	41.7 (38.3–45.3)	41.8 (28.5–56.5)	n.s.
Personal benefits: Taste good	39.3 (35.6–43.2)	39.7 (36.3–43.2)	38.0 (25.3–52.6)	n.s.
Initiation: Like the HTP technology	35.9 (32.2–39.7)	35.4 (32.1–38.9)	38.2 (25.8–52.3)	n.s.
**Initiation: The heating/charging device is attractive**	**33.6** **(30.1–37.1)**	**36.4** **(33.1–39.8)**	**20.0** **(11.8–31.6)**	**0.001**
Convenience: Use in places where smoking cigarettes are banned	31.3 (27.9–34.9)	32.8 (29.5–36.1)	24.4 (15.0–37.1)	n.s.
Personal benefits: Makes socializing easier	29.9 (26.5–33.5)	30.5 (27.4–33.8)	27.4 (17.2–40.7)	n.s.
Initiation: Offered by someone	26.7 (23.3–30.5)	25.8 (22.9–28.9)	31.5 (19.5–46.5)	n.s.
Personal benefits: Something to occupy time with	25.5 (22.4–28.9)	25.5 (22.6–28.7)	25.4 (15.4–38.0)	n.s.
**Product attractiveness: Attractive packaging**	**24.5** **(21.6–27.8)**	**26.6** **(23.6–29.7)**	**14.4** **(7.7–25.3)**	**0.007**
Initiation: People in the media or other public figures use HTPs	21.2 (18.2–24.6)	22.3 (19.4–25.4)	16.1 (8.1–29.7)	n.s.
Personal benefits: Save money	19.2 (16.1–22.3)	20.0 (17.4–22.9)	14.8 (7.4–27.4)	n.s.
Initiation: Experts like doctors and scientists use HTPs	17.7 (15.0–20.6)	19.1 (16.5–22.2)	10.2 (4.4–21.8)	n.s.
Personal benefits: Control appetite and/or weight	17.3 (14.7–20.2)	18.5 (15.9–21.4)	10.3 (4.6–21.3)	n.s
Initiation: Health professionals advised to switch to HTPs	16.9 (14.3–19.9)	17.4 (14.9–20.2)	14.1 (7.4–35.3)	n.s
Personal benefits: Look cool	15.7 (13.0–18.7)	16.0 (13.6–18.7)	14.2 (6.9–26.8)	n.s.
Intention: Cut down on the number of cigarettes I smoke	NA	38.2 (34.9–41.7)	NA	NA
Intention: Using HTPs might help me quit smoking cigarettes	NA	35.3 (32.0–38.8)	NA	NA
Intention: Replacing some cigarettes with HTPs to continue smoking	NA	25.3 (22.3–28.5)	NA	NA
**Quit–reduce composite measure**				
Using HTPs might help me to quit smoking cigarettes	NA	36.0 (32.6–39.4)	NA	NA
Using HTPs to reduce cigarette smoking, but not to quit smoking	NA	14.7 (12.2–17.3)	NA	NA
Using HTPs for other reasons (not to quit or reduce smoking cigarettes)	NA	49.3 (45.7–52.8)	NA	NA

Bolded cells are those where the difference between HTP-cigarette and exclusive HTP consumers is statistically significant at the *p* < 0.05 level; n.s.: not statistically significant; NA: not applicable.

## Data Availability

In each country participating in the International Tobacco Control Policy Evaluation (ITC) Project, the data are jointly owned by the lead researcher(s) in that country and the ITC Project at the University of Waterloo. Data from the ITC Project are available to approved researchers two years after the date of issuance of cleaned data sets by the ITC Data Management Centre. Researchers interested in using ITC data are required to apply for approval by submitting an International Tobacco Control Data Repository (ITCDR) request application and, subsequently, to sign an ITCDR Data Usage Agreement. The criteria for data usage approval and the contents of the Data Usage Agreement are described online (http://www.itcproject.org (accessed on 10 December 2022)).

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
