# Peer review of "Reasons for Initiation and Regular Use of Heated Tobacco Products among Current and Former Smokers in South Korea: Findings from the 2020 ITC Korea Survey"

_ijerph, 2023, doi:10.3390/ijerph20064963_

Round 1

Reviewer 1 Report

This manuscript aimed to explore the reasons for initiating and regularly using HTPs among exclusive, dual and triple HTP users. The topic of the manuscript is highly current as we still know very little about the motivations for HTP use.

In overall, this manuscript is well written. However, I have several minor comments that are listed below.

General comment

Please consistently use terminology either current or regular HTP use. I propose using ‘regularly using HTP’ or ‘regular HTP use’ because reasons for using HTPs regularly was assessed by the described measure.

Specific comments

Abstract

Line 32. Please revise: ‘…exclusive HTP first used HTP use and…’

Line 45. Please change ‘currently using’ to ‘regularly using’.

Lines 47-48. ‘…suggesting that HTPs in South Korea are not being conceptualized as an aid to smoking cessation.’ This was not emphasized in the Discussion. Please consider revising this part of the sentence.

Introduction

Line 57-58. ‘…that HTPs are lower in both compared to cigarettes,…’ Please revise, something maybe missing from this part.

Lines 85-86. Do you mean concurrently smoking cigarettes and/or using NVPs, and using HTPs? If yes, please clarify it in the sentence.

Lines 87-88. Please provide citation for the 'potential harm reduction benefit of HTPs'.

Line 93. ’HTP-cigarette consumer’: Please define briefly when first mentioned in the text. It is explained in the Methods section, but the Reader should understand the Introduction without reading the Methods.

Line 96. There is a double ‘that’.

Lines 96-97. Please consider changing this phrase as ‘…because they believe that HTPs are less harmful...’.

Methods

Line 128-130. Please consider writing ‘using both ≥weekly…’ instead of ‘both ≥weekly…’ and ‘using all three ≥weekly…’.

Line 144. ‘… (used HTPs and NVPs ≥weekly; …’ Perhaps a bracket is missing at the end, but do not repeat the description which is explained in line 130. I do not understand: ’ 117 were smoking cigarette occasionally (<weekly);’ do you mean out of 165 HTP users? Please clarify lines 144-145.

Lines 144-145. Why did you emphasize the ’stopped smoking completely’ group? Are they exclusive HTP users? If yes, explain this in lines 126-127. If there is a reason to separate them from exclusive HTP users, please provide a short explanation.

Line 162. ‘HTP Consumers’ In lines 126-130, different classification was used: 'exclusive HTP consumers'. Please use classification consistently. If exclusive HTP users were not included in the study, please revise the description of the sample by tobacco/nicotine product use.

Line 175. ‘In part derived by the study from…’ Please revise, perhaps something is missing here.

Table 1. Please consider changing the title to include in it that this table presents the outcome measure. E.g., ’Applied measure to assess reasons...’

Lines 206-215. I am confused about some analyses. Could you clearly indicate the results of ’separate adjusted logistic regression analyses (using the predicted marginal standardization method (PREDMARG)’, ’ …reasons stratified by smoking status (current vs. former/occasional smokers) to estimate reasons provided for HTP use within each group…’, and ’General linear contrasts of the predicted marginals in the corresponding models…’. Please clarify the existence of the results of these analyses in the manuscript.

Results

Line 232. Please change ‘Current Use of HTPs’ to ‘Regular Use of HTPs’.

Table 3. Please the title in the first column ‘Category/Reason’ to ‘Theme/Reason’.

Line 246-247. Please change the order of the following two reason: ‘…and they like the HTP technology (35.4%), attractive heating and charging devices (36.4%).’

Lines 249-250. Why not stress reduction (46.3%) was presented here in the text? Pleaase revise.

Line 262. Please use terms consistently. Previously it was always classified as 'HTP consumers' instead of ’Exclusive HTP Consumers’.

Line 265-266. ‘experts like doctor and scientists are using HTPs (19.1% vs 10.2%, p<0.05)’: In Table 3, it was indicated as not statistically significant. Please revise.

Line 267. Please change ’p<0.05’ to ’p<0.01’.

Line 268. Please revise percentages, different in the text like in Table 3. (for heating/charging devices are attractive)

Supplementary Table was not attached for review. Please consider not duplicating the results of the Supplementary Table 1 in the text.

Discussion

Line 301. Typing mistake: ’industy’

Line 308. Typing mistake: ’cigartete’

Line 314. Please correct to ’…one-fifth of Korean…’

Line 352. Typing mistake: ’curiocity’

Line 353-355. What could be in the background of these findings? Would you consider checking differences by socio-demographic characteristics of dual vs triple consumers?

Line 360. Typing mistake: non-combistible’

Conclusions

Please revise the main message of this part with the conclusions of the Abstract. These are somehow different. In the Abstract, the Authors wrote that ’only about one-third of HTP-cigarette consumers said they were using HTPs to quit smoking, suggesting that HTPs in South Korea are not being conceptualized as an aid to smoking cessation.’. The one-third could either high or low rate, but the reader cannot put these rate into a context. What about using other cessation methods? I the Authors would keep this line, please also discuss the frequency of using other cessation methods.

Supplementary Figure and Table were not attached for review so I could not review these.

Author Response

Thank you so much for your constructive feedback. I include all my responses to two reviewers in a single document. 

Reviewer 2 Report

The paper “Reasons for initiation and current use of heated tobacco products among HTP consumers in South Korea: Findings from the 2020 ITC Korea Survey” analyzed cross-sectional data from the 2020 International Tobacco Control (ITC) Korea Survey. The aim was to identify the reasons for Heated Tobacco Products (HTP) use (initiation and regular use). An authors’ specific interest was to investigate if cigarette consumers were using HTPs for smoking cessation or reduction. A secondary objective was to compare HTP-cigarette consumers and exclusive HTP consumers regarding reasons to use HTP.

The study presented is relevant regarding the rapid adoption of HTP by smokers (and by non-smokers???) and the benefits of this products for Public Health (comparing with tobacco smoking) that have been claimed by the tobacco industry. The study was well designed, was based in a large sample (representative of the population??) and used a proper data analysis strategy.

Results are interesting, are different of results from other countries (e.g., Japan) and have implications regarding the role of Public Health organizations in the shaping of reasons and decisions on the use of HTP by the users and the by the population (e.g., “as almost half of all HTP consumers use HTPs due to their perceived low harm, it is imperative to raise social awareness that HTP emissions are harmful to both consumers and non-consumers – l. 311-3). Overall, only 35.4% of HTP-cigarette consumers reported using HTPs to quit smoking, and only 14.7% to reduce smoking but not to quit. And, Overall, HTP consumers reported reasons to use HTP that are more related with convenience (less smelly than cigarettes - 71.3%) than with public or own health reasons (less harmful to own health than cigarettes - 48.6%). These are surprising results which contradict the tobacco industry communication about the “public health benefits” of using HTP.

I would like to highlight three conclusions of this study:

1) Most Korean HTP-cigarette consumers are using HTP to maintain their cigarette consumption rather than to quit.

2) HTPs may undermine tobacco control efforts since substantial numbers of Korean cigarette smokers use them for reasons other than to help quit smoking.

3) In the long term, HTP-cigarette use is unlikely to have any positive public health effects. This idea is in the discussion section but it is not in the conclusions section – I would like to suggest the inclusion of this idea in the conclusions section. 

In spite of this, some aspects of the manuscript should be clarified or revised. 

Abstract

- I do not understand this formulation: “… exclusive HTP first used HTP use and why …” (l. 32).

- Authors wrote: “165 [participants] were exclusive HTP consumers (using HTPs ≥ weekly) who were former or occasional smokers (smoking cigarette < weekly).” The information that these 165 exclusive HTP consumers were former or occasional smokers was confirmed or was assumed by authors? In different words, it is (or not) possible that some of these 165 exclusive HTP consumers started to use tobacco products by using HTP? The section 2.1. on Study Sample did not clarify this.

- Lines 36-8: “Respondents were asked to report the reason(s) they use HTPs, with 25 possible reasons for HTP-cigarette consumers, and 22 for exclusive HTP consumers.” This formulation suggests that the 22 possible reasons for exclusive HTP consumers were different of the 25 possible reasons for HTP-cigarette consumers … what is not the case. 

Methods

- Authors claimed that a representative sample was used in this study. This is an important claim that is not clear regarding the text. Is this a representative sample of South Korean adults? “We conducted a large population study among a representative sample of South Korean adults (l. 113)” or is this a representative sample of South Korean HTP consumers? “In this study, a nationally representative sample of South Korean HTP consumers (l. 286) … were examined…”. And this claim is not addressed in the methods section. 

- Once again, authors characterize the165 exclusive HTP consumers... “they had quit or were occasionally smoking...” (l. 143). Authors could confirm that no one of the exclusive HTP consumers started to use HTP without being a smoker before? 

- Regarding that one objective of this study was to investigate if cigarette consumers were using HTP for smoking cessation or reduction, some of the information that are in methods should be in results. For example (l. 144-5): about HTP exclusive consumers... 1) only 165 of 1815 participants (9%) were HTP exclusive consumers. Only 48 of 165 HTP consumers (29%) had stopped smoking completely. Only 22 of 165 HTP exclusive consumers (13%) were established former smokers. 

- Covariates: Regarding income, what do “low”, “moderate”, and “high” means? The same question for education. It is possible to get this information in Table 2, but this information must be presented in methods section. 

- The variable “geographic region” (l. 199) was not introduced in the covariates section. 

- Please consider rephrasing this text: “the authors categorize these reasons into six broad themes, i.e., 8 reasons examined HTP initiation and 14 reasons examined reasons for use among HTP consumers” (l. 175-6). 

- Reasons for initiating and regularly using HTP were categorized “into six broad themes” (l. 175) and further ahead in the text were mentioned as being categorized “under seven broad themes” (l. 364). This is not consistent. Please clarify. 

Results

- Table 2, Education High-HTP-Cigarette Consumer, %=86,6 (and not %=86,2). 

- Heading of Table 3: I suggest “Reasons” instead of “Themes of reasons” and “HTP consumers” instead of “those who were HTP consumers”. 

- 86% of HTP consumers had a high education level (university degree or higher). This is a remarkable figure. How does this figure compare with the general population? If HTP consumers are much more educated than the general population, this suggests that the adoption of HTP will increases considerably in the near future since high educated people tend to be behavior models for others. 

Discussion

- The text of the second paragraph (l. 295-301) is not based in the study results. Last sentence is related with results but is not clear the relation with the previous sentences. 

- This information must in the methods section (not here).: compared to existing studies, this study covers a variety of reasons for uisng HTPs under seven broad 364 themes: initiation, harm reduction, convenience, social consideration, product 365 attractiveness, personal benefits, and intention. 

- This information from the Discussion section is missing in the methods: “… this study covers a variety of reasons for using HTPs under seven broad 364 themes: initiation, harm reduction, convenience, social consideration, product 365 attractiveness, personal benefits, and intention.” 

- Again, this “(also ex- or occasional smokers)” is confirmed by the data of the study. See “… exclusive HTP consumers (also ex- or occasional smokers)” (l. 368). 

- The English of the text still needs a final revision – there are small errors like “curiocity” (l. 352), and, and “… a varity of reasons for uisng HTPs…” (l. 364).

Author Response

(The authors gave the same response as above.)
